# Adjuvant dendritic cell based immunotherapy (DCBI) after cytoreductive surgery (CRS) and hyperthermic intraperitoneal chemotherapy (HIPEC) for peritoneal mesothelioma, a phase II single centre open-label clinical trial: rationale and design of the MESOPEC trial

Nadine L de Boer,[1] Job P van Kooten,[1] Jacobus W A Burger,[1] Cornelis Verhoef,[1] Joachim G J V Aerts,[2] Eva V E Madsen[1]

JGJVA and EVEM contributed equally.

For numbered affiliations see end of article.

**Correspondence to**
Nadine L de Boer;
n.deboer@erasmusmc.nl

## ABSTRACT

**Introduction** Malignant peritoneal mesothelioma (MPM) is an uncommon but aggressive neoplasm and has a strong association with asbestos exposure. MPM has low survival rates of approximately 1 year even after (palliative) surgery and/or systemic chemotherapy. Recent advances in treatment strategies focusing on cytoreductive surgery (CRS) and hyperthermic intraperitoneal chemotherapy (HIPEC) have resulted in improved median survival of 53 months and a 5 year survival of 47%. However, recurrence rates are high. Current systemic chemotherapy in the adjuvant setting is of limited efficacy, while immunotherapy with dendritic cell based immunotherapy (DCBI) has yielded promising results in murine models with peritoneal mesothelioma and in patients with pleural mesothelioma.

**Methods and analysis** The MESOPEC trial is an open-label single centre phase II study. The study population are adult patients with histological/cytological confirmed diagnosis of epithelioid malignant peritoneal mesothelioma. Intervention: 4 to 6 weeks before CRS-HIPEC a leukapheresis is performed of which the monocytes are used for differentiation to dendritic cells (DCs). Autologous DCs pulsed with allogeneic tumour associated antigens (MesoPher) are re-injected 8 to 10 weeks after surgery, three times biweekly. Additional booster vaccinations are given at 3 and 6 months. Primary objective is to determine the feasibility of administering DCBI after CRS-HIPEC in patients with malignant peritoneal mesothelioma. Secondary objectives are to assess safety of DCBI in patients with peritoneal mesothelioma and determine whether a specific immunological response against the tumour occurs as a result of this adjuvant immunotherapy.

**Ethics and dissemination** Permission to carry out this study protocol has been granted by the Central Committee

### Strengths and limitations of this study

► The dendritic cell vaccines used in this protocol can be manufactured on a large scale, because autologous dendritic cells are loaded with allogeneic tumour associated antigens.

► Dendritic cell based immunotherapy has shown to have very limited side effects, especially when compared with systemic chemotherapy.

► This study will provide clinicians and scientists with important information about the immunological response after dendritic cell vaccination.

► Since all patients undergo cytoreductive surgery and hyperthermic intraperitoneal chemotherapy prior to dendritic cell based immunotherapy (DCBI), the effect of DCBI must be determined by assessing the immune response and overall clinical condition of each patient.

► In this phase II clinical trial the effect of DCBI on disease-free and overall survival cannot be determined, when DCBI is considered safe and feasible, a phase III clinical trial will be conducted to determine the effect on survival.

on Research Involving Human Subjects (CCMO in Dutch) and the Research Ethics Committee (METC in Dutch). The results of this trial will be submitted for publication in a peer-reviewed journal.

**Trial registration number** NTR7060. EudraCT: 2017-000897-12; Pre-Results.

## INTRODUCTION

Malignant peritoneal mesothelioma (MPM) is a highly lethal neoplasm, arising from the



serosal lining of the peritoneal cavity. It has a strong association with exposure to asbestos. Non-specific clinical symptoms like weight loss, abdominal pain and distension contribute to a delay in diagnosis. As a result, the majority of MPM cases are identified at an advanced stage, creating an overall poor life-expectancy of 4 to 12 months if left untreated.[1] Even after (palliative) surgery and/or systemic chemotherapy, MPM has poor survival of approximately 1 year.

In recent years treatment focus has shifted towards a more aggressive approach, utilising cytoreductive surgery (CRS) and hyperthermic intraperitoneal chemotherapy (HIPEC). Patients that underwent this treatment had a better prognosis with median survival of 53 months and 5 year survival of 47%.[2] However, even after CRS-HIPEC, recurrence rates are high with reported median progression-free survival and disease-free survival ranging from 11 to 28 months and 7.2 to 40 months respectively.[1] One explanation is that it is difficult to perform complete cytoreduction, as MPM often grows diffusely throughout the abdominal cavity,[3] but even when macroscopic complete cytoreduction is reached, loco-regional recurrence is often seen. A study that included 108 patients in whom complete or near-complete cytoreduction was achieved showed local recurrence in 38% of patients after median follow-up of 48.8 months.[4]

Effective adjuvant therapies are pressingly needed for above mentioned reasons. Dendritic cell based immunotherapy (DCBI) has shown promising results by harnessing the potency and specificity of the immune system. The first DCBI for mesothelioma was developed in the Erasmus MC, Rotterdam, and has been tested in murine models with peritoneal mesothelioma and in clinical phase I/II studies for patients with pleural mesothelioma.[5–11] These studies have shown that DCBI induces durable responses and higher survival rates compared with the general mesothelioma population. Dendritic cell (DC) therapy was well tolerated in these patients without grade 3 or 4 toxicities. Only low-grade fever and flu-like symptoms (grade 1 to 2) were seen for 24 hours after treatment. In a dose escalation phase I trial, the safety of using allogeneic tumour lysate (PheraLys) for the loading of the DCs was assessed. PheraLys is a tumour cell lysate derived from five well-characterised cell lines from patients with malignant mesothelioma. Tumour lysate priming strategies may be advantageous in providing the full antigenic repertoire of the tumour and might reduce the possibility of tumour escape by inducing a broader immune response. In this study adverse events were similar to earlier studies and importantly no severe adverse events were observed. Furthermore, clinical responses were established radiographically and some longtime survivors are being observed.

Previous preclinical studies demonstrated that DCBI has the capacity to slow down tumour growth, although tumour load has an important role in survival.[12] Mice had a better outcome when DCs were injected early in tumour development.[5] Mesothelioma cells produce specific cytokines and attract regulatory T-cells that suppress efficient immune responses, indicating that patients with low tumour load have a better functioning immune system and better anti-tumour responses.[13] Therefore it is the aim of this trial to treat patients with DCBI after complete macroscopic cytoreduction and HIPEC. The residual disease after cytoreductive surgery is classified using the the 'completeness of cytoreduction' (CCR score). CCR-0 indicates no visible residual tumour and CCR-1 indicates residual tumour nodules ≤2.5 mm. CCR-2 indicates residual tumour nodules between 2.5 mm and 2.5 cm. CCR-3 indicates a residual tumour >2.5 cm. In this study CCR ≤1 is considered as complete macroscopic cytoreduction. However, when complete cytoreduction cannot be achieved during surgery, patients undergo palliative HIPEC followed by DCBI

Main objective of this clinical trial is to determine feasibility of administering adjuvant DCBI after CRS-HIPEC. Secondary objectives are to asses safety and determine if a specific immunological response against the tumour occurs after DC therapy. When DCBI is considered safe and feasible as adjuvant treatment for patients with MPM, further research (phase III study) is warranted to determine the effect on survival.

## METHODS AND ANALYSIS
### Study design
#### Trial setting
The MESOPEC trial is an open-label, single arm, single centre phase II clinical trial. This study is conducted in the Erasmus MC, Rotterdam, an academic hospital located in the Netherlands. All patients included in this trial will receive adjuvant DCBI after CRS-HIPEC. Trial registration details are described in table 1.

#### Study population
The study population consists of adult patients diagnosed with MPM. Potentially eligible patients will be referred by their local clinician or through self-referral to a medical specialist and member of the study team to discuss the trial and determine eligibility.

In order to participate in the study, patients must meet the following inclusion criteria:
► Histologically or cytologically confirmed diagnosis of epithelioid MPM.
► WHO-Eastern Cooperative Oncology Group performance status of 0 or 1.
► Normal organ function and adequate bone marrow reserve (absolute neutrophil count >1.0×10⁹/l, platelet count >100×10⁹/l, haemoglobin >6.0 mmol/L).
► Positive delayed -type hypersensitivity skin test against at least one positive control antigen tetanus toxoid.
► Planned start date of vaccination within 8 to 10 weeks after CRS-HIPEC.
► Expected survival prior to surgery must be at least 6 months.

**Table 1** WHO trial registration data set

| Primary registry and trial identifying number | EudraCT number: 2017-000897-12<br>Netherlands trial register: NTR7060 |
|---|---|
| Date of registration in primary registry | October 2017 |
| Protocol version | Protocol version 3.0, date 31-10-2017 |
| SPIRIT guidelines data set for clinical trials | See online supplementary file |
| Source(s) of monetary or material support | Erasmus University Medical Centre, Rotterdam, the Netherlands<br>Dutch Cancer Society (KWF Kankerbestrijding)<br>Stichting Coolsingel |
| Primary sponsor | Erasmus University Medical Centre, Rotterdam, the Netherlands |
| Secondary sponsors | Dutch Cancer Society (KWF Kankerbestrijding)<br>Stichting Coolsingel |
| Contact for public queries | N.L. de Boer, study coordinator<br>Department of surgical oncology<br>Erasmus MC, Rotterdam, the Netherlands<br>n.deboer@erasmusmc.nl, (+31)010–704 21 25<br>J.P. van Kooten, study coordinator<br>Department of surgical oncology<br>Erasmus MC, Rotterdam, the Netherlands<br>j.kooten@erasmusmc.nl, (+31)010–704 21 25 |
| Contact for scientific queries | E.V.E. Madsen, principal investigator<br>Department of surgical oncology<br>Erasmus MC, Rotterdam, the Netherlands<br>e.madsen@erasmusmc.nl, (+31)010–704 10 82 |
| Public title | Adjuvant dendritic cell based immunotherapy (DCBI) after cytoreductive surgery (CRS) and hyperthermic intraperitoneal chemotherapy (HIPEC) for peritoneal mesothelioma: the MESOPEC trial |
| Scientific title | Adjuvant dendritic cell based immunotherapy (DCBI) after cytoreductive surgery (CRS) and hyperthermic intraperitoneal chemotherapy (HIPEC) for peritoneal mesothelioma: a phase II single centre open-label clinical trial |
| Countries of recruitment | The Netherlands |
| Health conditions or problems studied | Malignant peritoneal mesothelioma |
| Interventions | Vaccination with autologous dendritic cells loaded with allogeneic mesothelioma specific tumour antigens, after standard care (CRS-HIPEC) |
| Key inclusion and exclusion criteria | Inclusion criteria:<br>Confirmed diagnosis of epithelial peritoneal mesothelioma<br>WHO-ECOG performance status 0 to 1, expected survival at least 6 months<br>Adequate organ function and bone marrow reserves<br>Positive delayed type hypersensitivity skin test for positive control antigen<br>Exclusion criteria:<br>Extra-abdominal disease/metastatic disease<br>Current use of steroids or other immunosuppressive agents<br>Prior cytoreductive surgery<br>Prior malignancy other than basal cell carcinoma within 10 years of inclusion<br>Patients with a known allergy to shellfish<br>Serious chronic or acute illness considered to constitute unwarranted high risk for CRS-HIPEC or dendritic cell treatment<br>Pregnant or lactating women |
| Study type | Open label single centre phase II study |
| Date of first enrolment | March 2018 |
| Target sample size | 20 |
| Recruitment status | Recruiting |
| Primary outcome | Feasibility (DCBI therapy is considered feasible when 75% of patients enrolled in this study are able to receive and finish dendritic cell vaccination after CRS-HIPEC) |

Continued

de Boer NL, *et al. BMJ Open* 2019;**9**:e026779. doi:10.1136/bmjopen-2018-026779

| Table 1 | Continued | |
| --- | --- | --- |
| Primary registry and trial identifying number | EudraCT number: 2017-000897-12 Netherlands trial register: NTR7060 | |
| Key secondary outcome(s) | Safety Immunological response after dendritic cell vaccination | |

CRS, cytoreductive surgery; ECOG, Eastern Cooperative Oncology Group; HIPEC, hyperthermic intraperitoneal chemotherapy; SPIRIT, Standard Protocol Items: Recommendations for Interventional Trials.

► At least 18 years of age, written informed consent according to International Council for Harmonisation-Good clinical practice (ICH-GCP), ability to return to the study centre for adequate follow-up.

A potential participant who meets any of the following criteria will be excluded from participation in the study:

► Extra-abdominal mesothelioma (ie, metastatic disease).
► Prior cytoreductive surgery.
► Prior malignancy other than basal cell carcinoma within 10 years of inclusion.
► Serious concomitant disease or infection, including HIV or chronic viral hepatitis.
► Current use of steroids or other immunosuppressive agents (at least 6 weeks discontinuation before the first vaccine, with exception of prophylactic usage of dexamethasone during chemotherapy).
► History of auto-immune disease or organ allografts.
► Any disease that is considered to constitute an unwarranted high risk for CRS-HIPEC by the surgeon or study coordinator.
► Pregnant or lactating women.

### Patient timeline and additional procedures

Patients will undergo leukapheresis for dendritic cell vaccine production purposes 4 to 6 weeks before surgery. At baseline, subjects will undergo CRS-HIPEC.

At 6 weeks after surgery, the investigators will determine if the patient is sufficiently recovered and fit to undergo DCBI. Patients must have adequate bone marrow reserve before DCBI treatment; absolute neutrophil count $>1.0\times10^9$/l, platelet count $>100\times10^9$/l and haemoglobin >6.0 mmol/L.

Dendritic cell vaccinations will be given at 8 to 10 weeks after surgery three times biweekly. Before each vaccination laboratory testing will be performed and results reviewed before injection. Before and after injection vital signs (pulse, blood pressure, blood oxygen saturation and temperature) are determined. Patients are observed in the hospital for 2 hours after injection. Each vaccine contains at least $25\times10^6$ cells. One-third of this is injected intradermal, two-thirds are administered intravenous. Intradermal injection will be performed in the left arm. Intravenous injection will be performed via the vena brachialis in the left arm through a basic peripheral venous catheter. After the third vaccination, delayed-type hypersensitivity (DTH) skin test is performed. When DTH skin test result is positive for the dendritic cell vaccine, a 3 mm skin biopsy will be taken for further analyses. At 3 and 6 months after the first vaccination, two additional booster vaccinations will be given. Additional to study-related treatment, patients receive standard care and follow-up required after CRS-HIPEC.

Total duration of the treatment protocol is 8 to 9 months. In total, 11 additional visits are required to adhere to the study protocol. For the production of DCBI patients undergo leukapheresis. For the purpose of immune-monitoring, additional blood samples will be collected at seven moments during the study. (figure 1)

### Dendritic cell vaccine production

Dendritic cells are derived from peripheral blood mononuclear cells, by differentiating monocytes towards immature dendritic cells using specific cytokines. Immature dendritic cells are known for their high antigen uptake potential. Therefore they are exposed to tumour specific antigens in a co-culture with allogeneic mesothelioma cell lysate. This lysate (PheraLys) is derived from five well-specified mesothelioma cell lines. After exposure to PheraLys the immature dendritic cells are differentiated towards mature dendritic cells, which are ready to activate the immune system in vivo (MesoPher). (figure 2)

### Withdrawal of individual subjects

Subjects can leave the study at any time for any reason if they wish to do so without any consequences. The investigator can decide to withdraw a subject from the study for urgent medical reasons. Should a participant withdraw from the trial, then every effort will be made to obtain follow-up data, with the permission of the patient.

The investigators also have the right to withdraw patients from the study if one or more of the following events occur:

► Significant protocol violation or non-compliance on the part of the patient or investigator.
► Refusal of the patient to continue treatment or observations.
► Any change in the condition of the patient that justifies discontinuation of treatment.
► Decision by the study coordinator that MesoPher does not comply to quality requirements (advice of qualified person).
► Decision by the study coordinator that termination is in the patient's best medical interest.
► Unrelated medical illness or complication.

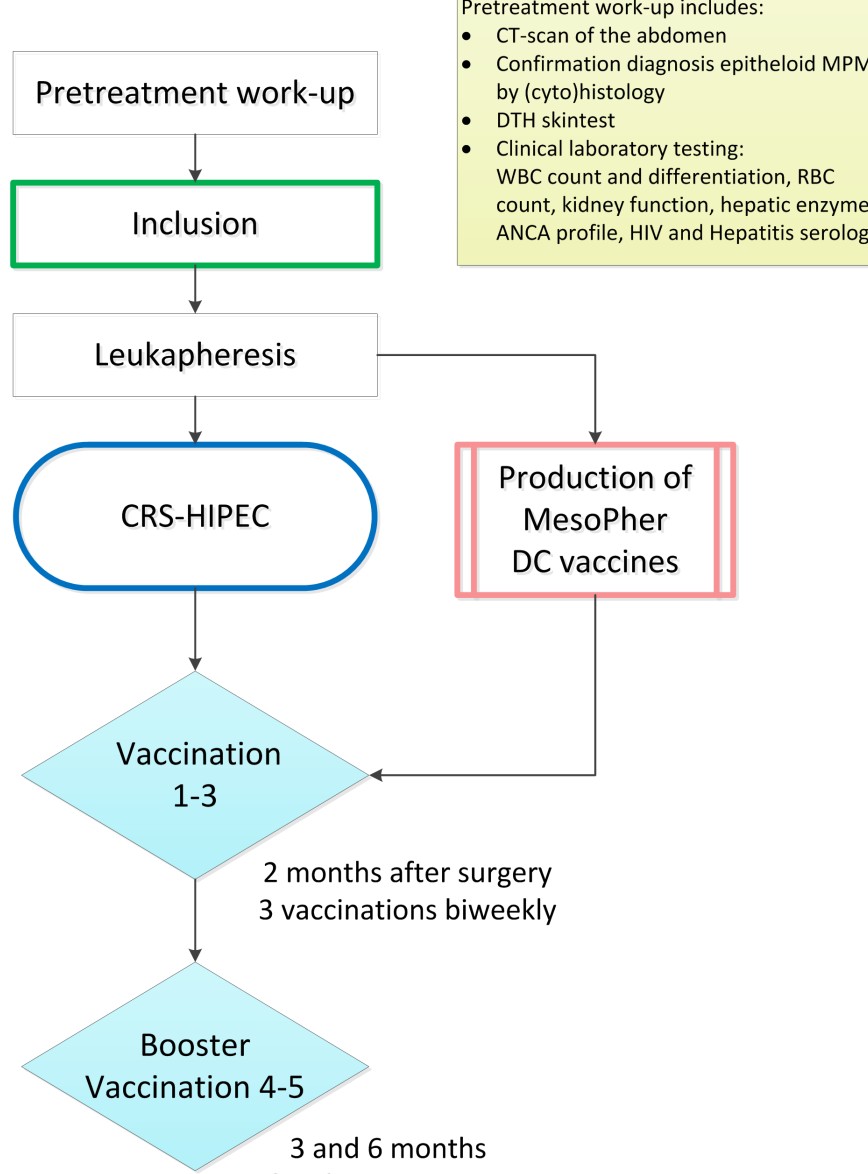

Pretreatment work-up includes:
- CT-scan of the abdomen
- Confirmation diagnosis epitheloid MPM by (cyto)histology
- DTH skintest
- Clinical laboratory testing: WBC count and differentiation, RBC count, kidney function, hepatic enzymes, ANCA profile, HIV and Hepatitis serology.

**Figure 1** Patient timeline: In total 11 additional visits are required. After informed consent is acquired, screening will take place in the form of full examination and DTH skin test. When patients comply to all criteria, they will undergo leukapheresis for production of dendritic cell vaccine. After 2 to 4 weeks patients undergo CRS-HIPEC. First vaccination is given 8 to 10 weeks after surgery, followed by two more vaccinations biweekly. DTH skin testing is performed for analysis of immune response 2 weeks after the third vaccination. At 3 and 6 months after first vaccination, subjects receive additional 'booster' vaccination. Each vaccine contains at least $25\times10^6$ cells. One-third is injected intradermal. Two-thirds are administered intravenous. ANCA, anti-neutrophil cytoplasmic antibody; CRS, cytoreductive surgery; CT, computed tomography; DC, dendritic cell; DTH, delayed-type hypersensitivity; HIV, human immunodeficiency virus; HIPEC, hyperthermic intraperitoneal chemotherapy; MPM, malignant peritoneal mesothelioma; RBC, red blood cell; WBC, white blood cell.

► Serious logistical problems of practical problems in clean room.

## Objectives and analysis
### Primary objective
Primary objective is to determine feasibility of using DCBI after CRS-HIPEC in patients with peritoneal mesothelioma. DCBI after CRS-HIPEC is deemed feasible when at least 75% of patients that are included in this trial complete the full treatment schedule. This cut-off is based on the fact that currently around 75% of patients undergoing CRS-HIPEC are fit to undergo adjuvant therapy, such as systemic chemotherapy.

### Secondary objectives
Secondary objectives are to assess safety of DCBI therapy after CRS-HIPEC and determine whether a specific immunological response occurs due to dendritic cell vaccination.

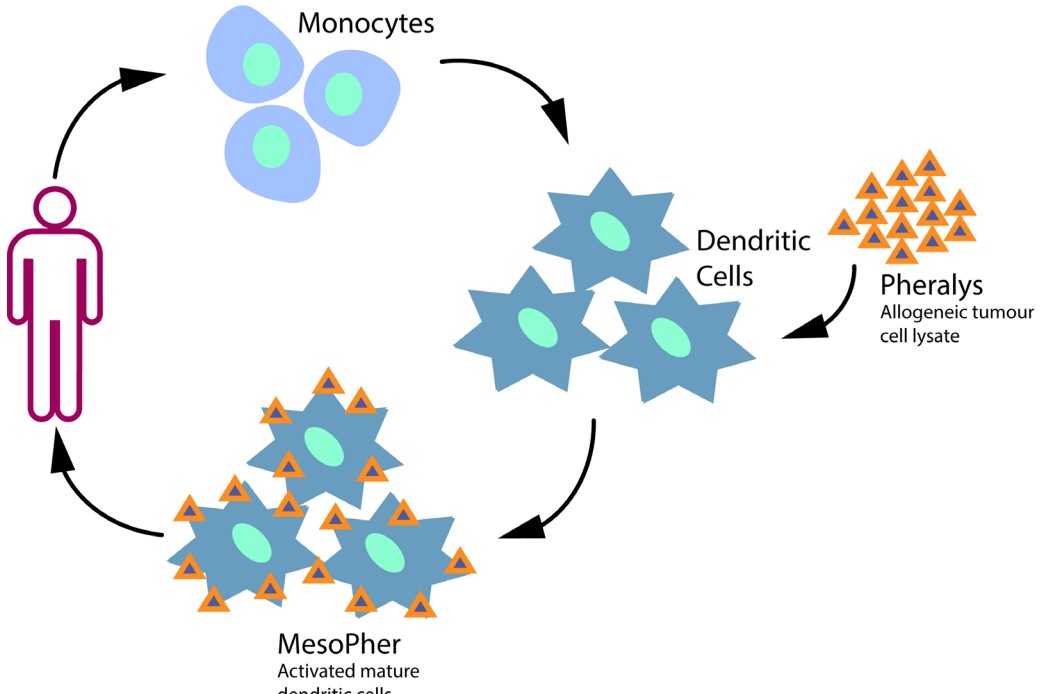

**Figure 2** DCBI production process: Monocytes are isolated from peripheral blood and are then stimulated to differentiate towards immature dendritic cells. These immature dendritic cells are exposed to PheraLys tumour cell lysate. After further differentiation towards mature dendritic cells, MesoPher vaccinations are given back to the patient. DCBI, dendritic cell based immunotherapy.

## Safety

Previous clinical studies have shown that injection with tumour lysate-pulsed autologous DCs was overall well tolerated without systemic toxicity, with the exception of low-grade flu-like symptoms like fever and rigours. No grade 3 or 4 toxicity was observed.[11] However, safety and tolerability after major surgery has yet to be determined.

The administration of autologous cells that have been loaded with allogeneic human materials is a potential health risk. Because not the lysate itself is administered to the patients, but only after it has been processed by autologous dendritic cells, risks are expected to be limited.

The necessary sample size for the detection of grade 3 toxicity is calculated to be 20 patients (figure 3). All adverse events, serious adverse events and suspected unexpected serious adverse reactions will be monitored and reported by the sponsor.

## Immune response

Assessment of immune responses will be conducted on three levels in all treated patients; (1) responses that mark successful vaccination, (2) enhanced frequencies

$$N = z_{0.975}^2 \cdot \frac{sens(1 - sens)}{w^2 \cdot prev}$$

**Figure 3** Sample size calculation: Assuming the sensitivity for detecting grade 3 (or higher) toxicity is 99%. Expected prevalence of grade 3 toxicity in the study population is 2.5%.

of tumor-specific T cells in peripheral blood samples and (3) frequency shifts in other immune cell subsets.

1. Keyhole limpet hemocyanin (KLH) is added to the vaccines as a surrogate marker. With the use of KLH, we will assess whether an immune response against the vaccine has occurred and whether this response persists. KLH is known to induce a specific adaptive immune response readily detectable in sera (antibody response) and skin tests (cellular response) of vaccinated individuals. Serum samples will be collected before, during and after DCBI as well as at selected intervals during follow-up. Humoral responses to KLH will be detected using ELISA. Furthermore, patients will undergo a DTH skin test before and after DCBI. DTH responses will be evaluated for local inflammation after 48 hours and 3 mm punch biopsies will be collected in case inflammation occurs. These biopsies will subsequently be used for in situ immunostainings of DC, myeloid derived suppressor cells (MDSC), and CD8 +T cells.

2. To assess vaccine-induced frequencies of tumor-specific T cells, we will conduct a potency assay in accordance with the 'Guideline on potency testing of cell based immunotherapy medicinal products for the treatment of cancer' as provided by the European Medicine Agency (EMA) in 2016 (EMA/CHMP/BWP/271475/2006 rev.1). The proposed assay can shortly be described as a co-culture of T cells isolated from pre- and post-treatment peripheral blood samples with autologous DCs loaded with autologous tumour lysate. Subsequent-

ly, we will measure T cell proliferation and activation markers via multicolour flow cytometry.

3. Phenotypical analysis of immune cell subsets will be conducted using flow cytometry to detect vaccine-induced changes in the frequencies of >100 immune cell (subsets) that represent distinct lineages and/or express different levels of activation, differentiation and co-signalling markers. Staining of fresh patient material at different time points will allow enumeration of different immune cells throughout therapy. Subsequent bulk analysis of frozen material focusses solely on an extensive array of T cell, MDSC and DC markers. Combination of these readouts allows for the generation of immune profiles for individual patients. The analysis of these profiles in turn will allow for the determination of prospective markers and better stratification of patient populations suited for vaccination therapy.

## Statistical analysis

The statistical analyses/data summaries will be performed using SPSS. Other tools may be used for exploratory summaries and graphical presentations.

Primary endpoint is to determine the feasibility of administering DCBI after CRS-HIPEC in patients with malignant peritoneal mesothelioma. Based on previous results it is known that from all patients with colorectal carcinoma that underwent CRS-HIPEC, about 25% is not able to receive adjuvant systemic therapy due to postoperative complications. Feasibility is set if 15 of 20 patients (75%) were able to undergo leukapheresis successfully, production of PheraLys and dendritic cell vaccines was successful and when patients were able to complete the full adjuvant treatment schedule.

## Patient and public involvement

The Dutch patient association for patients with mesothelioma 'Instituut Asbestslachtoffers' and the Erasmus MC, Rotterdam, have worked together closely for years on the development of dendritic cell immunotherapy for mesothelioma. The patient association has received a copy of the study protocol to comment on the research question and outcome measures and also received the patient information folder to comment on patients preference and clarity of the information folder. During the study, feedback is provided on the inclusion rate and patient experiences. The results of the study will also be communicated to the patient association, which can then distribute them among their members.

## ETHICS AND DISSEMINATION

The study will be conducted in compliance with the 'Medical Research Involving Human Subjects Act' (WMO) and according to the principles of the Declaration of Helsinki (64th World Medical Association General Assembly, Fortaleza, Brazil, October 2013).

If protocol modifications occur, the new protocol has to be approved by the Central Committee on Research Involving Human Subjects and the Research Ethics Committee of the Erasmus MC before they can be implemented. Data collection, data assessment and data analysis will be performed according to the local guidelines for data management of the Erasmus MC.

To generate more awareness for this current study and to increase referrals of potential study candidates to the Erasmus MC, a short Dutch summary of the study will be submitted to The Dutch Journal for Oncology (NTVO in Dutch). Also, presentations about the study have been given at the Dutch Society of Surgery meeting and the Peritoneal Surface Oncology Group International meeting in Paris.

The results of this clinical trial will be submitted for publication in a peer-reviewed scientific journal.

The investigator will permit auditors to carry out site visits to audit the compliance with regulatory guidelines. Similar auditing procedures may be conducted by agents of any regulatory body reviewing the results of this study.

The sponsor will submit a yearly safety report to the accredited METC, competent authority and competent authorities of the concerned Member States. This safety report consists of; 1 a list of all suspected (unexpected or expected) serious adverse reactions, 2 a report concerning the safety of the subjects consisting of a complete safety analysis and an evaluation of the balance between the efficacy and the harmfulness of the medicine under investigation.

Within 1 year after the end of the study, the investigator/sponsor will submit a final study report with the results of the study, including any publications/abstracts of the study, to the accredited METC and the Competent Authority.

Currently three patients are included in the study protocol. The final patient is expected to be included at the end of 2020. First results are expected in 2021.

## DISCUSSION

The main objective of the MESOPEC trial is to determine feasibility of DCBI as adjuvant treatment after CRS-HIPEC in patients with MPM. Secondary objectives are to assess safety and to monitor the immune response after DCBI. The MESOPEC trial is the first clinical trial offering adjuvant dendritic cell immunotherapy to patients with MPM. So far (neo)adjuvant systemic chemotherapy has shown no benefit on surgical or oncological outcome for MPM.[14] Systemic chemotherapy, using cisplatin and pemetrexed, is standard treatment in pleural mesothelioma and has been applied to patients with peritoneal mesothelioma. This has shown limited efficacy, considerable toxicity and even mortality, making it unfit for the treatment of MPM.[15]

The DCBI used in this current trial consists of personalised dendritic cell vaccines, produced with autologous dendritic cells that are loaded with tumour associated

antigens derived from allogeneic tumour cell lysate. This treatment approach has multiple advantages. The biggest advantage of this strategy is that it is possible to produce personalised anti-cancer vaccines on a scale sufficient for clinical implementation in larger groups of patients. Another advantage is that so far DCBI has shown no severe side effects and therefore causes little morbidity especially when compared with current other adjuvant treatment options, like systemic chemotherapy.

Analyses of tissue and blood samples that are collected throughout the study will provide valuable information for scientists and clinicians regarding the immunologic response after DCBI. Furthermore, potential of DCBI can be tested in vitro by culturing tumour cell lines derived from tumour tissue obtained during CRS. By doing so, in the future, patients that will respond to immunotherapy can be identified before starting treatment.

Currently, CRS-HIPEC is the gold standard for a selected group of patients suffering from MPM. Eligibility for CRS-HIPEC is dependent on several clinical aspects such as 'peritoneal carcinoma index' (PCI), histological subtype and overall patient health condition. Previous studies have shown that outcome after CRS-HIPEC strongly depends on the completeness of cytoreduction.[1] Unfortunately it is very difficult to achieve complete cytoreduction in patients with high PCI score. It has been reported by others that immunotherapy is possibly more effective in patients with low tumour load. Therefore, in this study DCBI is given as adjuvant treatment after CRS-HIPEC. However, if a significant clinical effect can be achieved, in the future DCBI might be used as a neoadjuvant therapy making CRS-HIPEC with complete cytoreduction available for a larger number of patients.

We acknowledge the fact that this study has limitations. One limitation of this study is the small number of patients that will be included. Given the rarity of MPM, it is difficult to include large numbers of patients. However the sample size of this current study, should be sufficient to determine the feasibility and safety of adjuvant DCBI after CRS-HIPEC.

Another limitation of this study design is that it is not possible to determine radiological response to DCBI treatment. After debulking surgery, MPM will not be detectable on CAT scans. Therefore the response to DCBI treatment must be determined by assessing the immune response and overall clinical condition of each patient. Clinical effect of DCBI on overall survival can only be determined after a longer period of follow-up.

If DCBI is considered feasible as adjuvant treatment for MPM, a larger phase III clinical trial should be conducted to determine the effect on surgical and oncological outcome. Because MPM incidence in the Netherlands alone is very low, this clinical trial would have to be conducted internationally.

**Author affiliations**
¹Surgical Oncology, Erasmus MC, Rotterdam, Zuid-Holland, The Netherlands
²Pulmonary Medicine, Erasmus MC, Rotterdam, Zuid-Holland, The Netherlands

**Correction notice** This article has been corrected since it first published online. The open access licence type has been amended.

**Acknowledgements** We would like to thank the Dutch patient association for patients with mesothelioma for their detailed advice and guidance during the design of this study.

**Contributors** NLdB and JPvK drafted this manuscript. NLdB, JWAB, JGJVA, CV and EVEM drafted the original study protocol and revised the manuscript. CV, JGJVA and EVEM initiated the trial and supervised the drafting of the study protocol and manuscript. EVEM acquired funding for implementation of the trial protocol and is the primary clinical investigator. All authors approved the final manuscript.

**Funding** This work was supported by: The Dutch Cancer Society (in Dutch: KWF Kankerbestrijding), grant number: 10246, Stichting Coolsingel, grant number: 482

**Competing interests** None declared.

**Patient consent for publication** Not required.

**Provenance and peer review** Not commissioned; externally peer reviewed.

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
