## [Reviewer comments · BMJ Open]

ARTICLE DETAILS

TITLE (PROVISIONAL)	Adjuvant dendritic cell based immunotherapy (DCBI) after cytoreductive surgery (CRS) and hyperthermic intraperitoneal chemotherapy (HIPEC) for peritoneal mesothelioma, a phase II single center open-label clinical trial. Rationale and design of the MESOPEC trial.
AUTHORS	de Boer, Nadine; van Kooten, Job; Burger, Jacobus; Verhoef, Cornelis; Aerts, Joachim; Madsen, Eva

VERSION 1 - REVIEW

REVIEWER	Pocard Marc INSERM Paris 7 University France
REVIEW RETURNED	03-Nov-2018

GENERAL COMMENTS	Of course no major change can be done in the methodology however I have only on point to offer as a simple question, requiring a simple answer and a second more complex but maybe important. i) Patients will require, sometime, to obtain a complete cytoreductive surgery a splenectomy. That resection can change the result of immune treatment. That information had to be recorded and the fact that inclusion in the protocol will be possible even after splenectomy had to be noticed in the protocol. ii) More complex: A sort of accepted clinical and biological base line at 6 postoperative weeks to receive the treatment had to be more define than "it is possible" OMS status, with blood cells number, presence or not of a limited infection? That could help because at 6 weeks for some patients, including if the receive preoperative chemotherapy using anti-angiogenic drugs, wound healing could be incomplete and it is better to define all situations to have results "feasibility of a phase 2" strong enough to offer a phase 3 access.
--

REVIEWER	Chukwuemeka Ihemelandu MedStar Georgetown University Medical Center
REVIEW RETURNED	05-Nov-2018

GENERAL COMMENTS	I applaud the authors on this bold attempt to address the very pressing issue of the need for more effective therapeutics for the treatment of peritoneal mesothelioma
--

REVIEWER	MICHAEL KLUGER Michael D. Kluger, MD, MPH Assistant Professor of Surgery Division of GI & Endocrine Surgery Columbia University College of Physicians and Surgeons New York-Presbyterian Hospital 161 Fort Washington Ave - 8th Floor New York, New York 10032
REVIEW RETURNED	10-Dec-2018

GENERAL COMMENTS	pg 7 line 31 should be above mentioned pg 12 line 52 should be gold not golden Please comment on HIPEC protocol. "Mice had a better outcome when DCs were injected early in tumor development." How early, and will the 6-8 week delay in this study be an issue? "Therefore in this trial DCBI is given as an adjuvant treatment after complete macroscopic cytoreduction and HIPEC." How is this being defined?
---

REVIEWER	Maximiliano Gelli Gustave Roussy Cancer Campus (France)
REVIEW RETURNED	20-Jan-2019

GENERAL COMMENTS	This is an interesting phase II study evaluating feasibility of DCBI after CRS + HIPEC defined according the the possibility to perform dendritic cell vaccination in three quarter of patients. Unfortunately no clear criteria to administrate the vaccination are mentioned in the manuscript that could arise interobserver variability. if these criteria are present in the protocol, please include in the protocol. Adjuvant chemotherapy can be sometimes performed after aggressive CRS and HIPEC MP. How can you integrate adjuvant treatment in this context?
--

VERSION 1 – AUTHOR RESPONSE

Reviewer(s) Reports:

Reviewer: 1

Reviewer Name: Pocard Marc

Institution and Country: INSERM, Paris 7 University, France

Please leave your comments for the authors below:

Of course no major change can be done in the methodology however I have only on point to offer as a simple question, requiring a simple answer and a second more complex but maybe important.

1. Patients will require, sometime, to obtain a complete cytoreductive surgery a splenectomy. That resection can change the result of immune treatment. That information had to be recorded and the fact that inclusion in the protocol will be possible even after splenectomy had to be noticed in the protocol.

Thank you for your comment. We agree with the reviewer that a splenectomy can change the result of immune treatment. In the phase III trial , following this phase II trial, this should definitely be mentioned in the study protocol. However, in this current study, our primary goal is to establish “feasibility“ of this treatment. We do not believe that the feasibility of immune treatment is influenced by a splenectomy. Therefore we did not mention this in our in-/exclusion criteria. Nevertheless, in the current study we will add the variable ‘splenectomy’ to our database, and include this in our analysis.

2. More complex: A sort of accepted clinical and biological base line at 6 postoperative weeks to receive the treatment had to be more define than “it is possible” OMS status, with blood cells number, presence or not of a limited infection? That could help because at 6 weeks for some patients, including if the receive preoperative chemotherapy using anti-angiogenic drugs, wound healing could be incomplete and it is better to define all situations to have results “feasibility of a phase 2” strong enough to offer a phase 3 access.

Thank you for your comment we added the following text to our manuscript (page 6);

‘At six weeks after surgery, the investigators will determine if the patient is sufficiently recovered and fit to undergo DCBI. Patients must have adequate bone marrow reserve before DCBI treatment: absolute neutrophil count $>1.0 \times 10^9/l$, platelet count $>100 \times 10^9/l$ and Hb $>6.0\text{mmol/l}$. Dendritic cell vaccinations will be given at eight to ten weeks after surgery three times biweekly. Before each vaccination laboratory testing will be performed and results reviewed before injection. Before and after injection vital signs (pulse, blood pressure, blood oxygen saturation and temperature) are determined. Patients are observed in the hospital for two hours after injection.’

Reviewer: 2

Reviewer Name: Chukwuemeka Ihemelandu

Institution and Country: MedStar Georgetown University Medical Center

Please leave your comments for the authors below :

I applaud the authors on this bold attempt to address the very pressing issue of the need for more effective therapeutics for the treatment of peritoneal mesothelioma We would like to thank reviewer 2 for his kind words.

Reviewer: 3

Reviewer Name: Michael Kluger

Institution and Country: Assistant Professor of Surgery, Division of GI & Endocrine Surgery, Columbia University College of Physicians and Surgeons, New York-Presbyterian Hospital, 161 Fort Washington Ave - 8th Floor New York, New York 10032

Please leave your comments for the authors below :

1. pg 7 line 31 should be above mentioned

Thank you for your comment, the grammatical error has been adjusted.

2. pg 12 line 52 should be gold not golden

“Golden standard“ is changed to “gold standard” in the revised manuscript.

Please comment on HIPEC protocol:

3. "Mice had a better outcome when DCs were injected early in tumor development." How early, and will the 6-8 week delay in this study be an issue?

In the study we refer to (reference 5, Hegmans et al, Am J Respir Care Med, 2005) mice were treated with DCBI after 1, 3 and 5 days after lethal tumor dose was injected intraperitoneally. Mice injected with DCBI early in tumor development showed better survival. This study, as well as others, suggests that patients with lower tumor load show better anti-tumor responses. In this current study patients are treated with DCBI 6-8 weeks after CRS-HIPEC. We are not concerned with the six to eight week post-operative delay. Though peritoneal mesothelioma is characterized by its aggressive growth, six to eight weeks after CRS-HIPEC, tumor load will still be low. Besides that; patients need time fully to recover from CRS-HIPEC treatment before they start with DCBI treatment. This usually takes 4-6 weeks.

4. "Therefore in this trial DCBI is given as an adjuvant treatment after complete macroscopic cytoreduction and HIPEC." How is this being defined?

Thank you for your comment, we added a definition of complete macroscopic cytoreduction to page 5:

'Therefore it is the aim of this trial to treat patients with DCBI after complete macroscopic cytoreduction and HIPEC. The residual disease after cytoreductive surgery is classified using the the 'completeness of cytoreduction' (CCR score). CCR-0 indicates no visible residual tumor and CCR-1 indicates residual tumor nodules ≤ 2.5 mm. CCR-2 indicates residual tumor nodules between 2.5 mm and 2.5 cm. CCR-3 indicates a residual tumor > 2.5 cm. In this study $CCR \leq 1$ is considered as complete macroscopic cytoreduction. However, when complete cytoreduction cannot be achieved during surgery, patients undergo palliative HIPEC followed by DCBI.'

Reviewer: 4

Reviewer Name: Maximiliano Gelli

Institution and Country: Gustave Roussy Cancer Campus (France)

Please leave your comments for the authors below:

1. This is an interesting phase II study evaluating feasibility of DCBI after CRS + HIPEC defined according to the possibility to perform dendritic cell vaccination in three quarter of patients. Unfortunately no clear criteria to administrate the vaccination are mentioned in the manuscript that could arise interobserver variability. If these criteria are present in the protocol, please include in the protocol. Thank you for your comment. The instructions for administration of the vaccination are briefly mentioned in Figure 1. One third is injected intradermal. Two thirds are administered intravenous. As has been done in our previous trial (Aerts et al, Clin Cancer Res, 2018) and in the ongoing phase III study. Besides that, since this is a single center study all vaccinations are administered by the same study team, which were all trained for this procedure. However, we agree with reviewer 4 that the criteria for administration need to be clear and these are therefore added to section 2.1.3 on page 6: 'Each vaccine contains at least 25×10^6 cells. One third of cells are injected intradermal, two thirds are administered intravenous. Intradermal injection will be performed in the

upper left arm. Intravenous injection will be performed via the vena brachialis in the left arm through a basic peripheral venous catheter.'

2. Adjuvant chemotherapy can be sometimes performed after aggressive CRS and HIPEC MP. How can you integrate adjuvant treatment in this context?

Thank you for your comment. In this current study patients will not receive adjuvant chemotherapy after CRS-HIPEC. This study is performed specifically to determine the feasibility of adjuvant DCBI, without the toxicity or interference of systemic chemotherapy. Thereby, earlier studies (for example; Deraco et al. Ann Surg Oncol, 2013) have suggested that adjuvant chemotherapy has no significant impact on survival after CRS-HIPEC for MPM. This is one of the main incentives for conducting this trial.

VERSION 2 – REVIEW

REVIEWER	Pocard, M INSERM France
REVIEW RETURNED	09-Feb-2019

GENERAL COMMENTS	the recent version respond to the reviewer's question
---

REVIEWER	Gelli, Maximiliano Gustave Roussy Grance
REVIEW RETURNED	22-Feb-2019

GENERAL COMMENTS	Authors took into account the comments
--